# EmbeddedPigCount: Pig Counting with Video Object Detection and Tracking on an Embedded Board

**DOI:** 10.3390/s22072689

**Published:** 2022-03-31

**Authors:** Jonggwan Kim, Yooil Suh, Junhee Lee, Heechan Chae, Hanse Ahn, Yongwha Chung, Daihee Park

**Affiliations:** 1Info Valley Korea Co., Ltd., Anyang-si 14067, Korea; jking@invako.kr (J.K.); yoor0815@invako.kr (Y.S.); lyjourney@invako.kr (J.L.); chai@invako.kr (H.C.); 2Department of Computer Convergence Software, Korea University, Sejong 30019, Korea; hansahn@korea.ac.kr (H.A.); dhpark@korea.ac.kr (D.P.)

**Keywords:** agriculture IT, computer vision, pig counting, video object detection and tracking, convolutional neural network

## Abstract

Knowing the number of pigs on a large-scale pig farm is an important issue for efficient farm management. However, counting the number of pigs accurately is difficult for humans because pigs do not obediently stop or slow down for counting. In this study, we propose a camera-based automatic method to count the number of pigs passing through a counting zone. That is, using a camera in a hallway, our deep-learning-based video object detection and tracking method analyzes video streams and counts the number of pigs passing through the counting zone. Furthermore, to execute the counting method in real time on a low-cost embedded board, we consider the tradeoff between accuracy and execution time, which has not yet been reported for pig counting. Our experimental results on an NVIDIA Jetson Nano embedded board show that this “light-weight” method is effective for counting the passing-through pigs, in terms of both accuracy (i.e., 99.44%) and execution time (i.e., real-time execution), even when some pigs pass through the counting zone back and forth.

## 1. Introduction

Pork is one of the most consumed meats in the world, and the average consumption of pork annually is approximately 1.05 million tons globally. Moreover, 1.08 million tons of pork was delivered to customers in 2020, and the pork meat market size is projected to reach 17% from 2021 to 2029 (OECD 2021) [1]. With the increase in pork meat demand, piggery farms also need to expand to meet these needs. This leads to an increase in the number of pigs that each pig farmer must take care of on the farms. If the number of pigs that one worker manages increases, piggery farms may face countless human errors. In other words, farms possibly lose track of pigs and frequently become unmanageable. Pig tracking and counting on pig farms is an essential part of pig management, and accurate pig counting automatically promotes efficient management and reduces manpower input for inspection.

Over the last few years, some monitoring techniques have been extensively applied to livestock farming [2,3,4,5,6], and several studies have utilized surveillance systems to monitor pigs automatically [7,8,9,10]. The aim of this study is to analyze video-based pig monitoring, using non-attached (i.e., non-invasive) sensors [11,12,13,14,15,16,17,18,19,20,21,22,23,24,25,26,27,28,29,30,31,32,33,34,35,36,37,38]. Moreover, we adopt a top-view camera [18,19,20,21,22,23] to resolve general issues, such as occlusion, overlapping, illumination changes, and rapid movements during pig monitoring. Recently, end-to-end deep learning techniques have been widely used for computer vision applications (i.e., object recognition, object classification, and object detection), but these deep learning techniques require large numbers of parameters and, thus, high computational costs. To apply deep learning techniques to video-based pig counting, we must process each video frame in real time from a video stream, without any delay.

In this study, we focus on real-time pig counting, using an embedded board for low-cost monitoring. For a large-scale pig farm, practical issues, such as monitoring costs, should be considered. For example, owing to the severe ammonia gas in a pig room, any PCB board will be corroded faster than normal monitoring environments; thus, a low-cost solution is required for the practical monitoring of a pig room. However, executing typical deep learning techniques on an embedded board cannot satisfy the real-time requirements for video monitoring. Therefore, we focused on developing a method for meeting both accuracy and real-time requirements, with a low-cost embedded board. The contributions of the proposed method are summarized as follows.
For intelligent pig monitoring applications with low-cost embedded boards, such as the NVIDIA Jetson Nano [39], light-weight object detection and tracking algorithms are proposed. By reducing the computational cost in TinyYOLOv4 [40] and DeepSORT [41], we can detect and track pigs in real time on an embedded board, without losing accuracy.An accurate and real-time pig-counting algorithm is proposed. Although the accuracies of light-weight object detection and tracking algorithms are not perfect, we can obtain a counting accuracy of 99.44%, even when some pigs pass through the counting zone back and forth. Furthermore, all counting steps can be executed at 30 frames per second (FPS) on an embedded board. To the best of our knowledge, the trade-off between execution time and accuracy in pig counting on an embedded board has not been reported.

The remainder of this paper is organized as follows: Section 2 summarizes previous methods for pig detection and/or counting. Section 3 describes the proposed method for detecting, tracking, and counting pigs. Section 4 presents the experimental results, and Section 5 concludes the paper.

## 2. Background

The final goal of this study is to track and count pigs that walk in a hallway with embedded boards in a cost-effective manner. Accurate results from detection models are required for reliable pig tracking and counting. Most previous studies detected pigs using image [11,12,13,14,15,16,17,18,19,20,21,22,23,24,25,26] and video [27,28,29,30,31,32,33] object detection techniques. The majorities of recent methods utilize end-to-end deep learning techniques for object detection problems, and convolutional neural networks (CNNs) are the most frequently used solutions to provide stable and accurate results for object detection. CNNs for object detection can be categorized into two groups: two-stage and one-stage detectors. Two-stage detectors, such as R-CNN [42], fast R-CNN [43], and faster R-CNN [44], use two networks to process regional proposal and classification. By contrast, you only look once (YOLO) [45] and single shot multibox detector (SSD) [46] are one-stage detectors that use one network to handle regional proposal and classification simultaneously. Typically, two-stage detectors show more accurate localization than one-stage detectors. However, two-stage detectors incur high computational costs because they contain numerous parameters. In summary, faster R-CNN is slightly more accurate than YOLO, but YOLO is much faster than faster R-CNN. Thus, we will modify TinyYOLOv4 [40], which is a tiny version of YOLOv4 [40], to deploy it on embedded boards.

Accurate pig tracking is also required for pig counting. Because pigs do not take a pose for counting, pig counting with videos rather than images is required. In addition, video has been used for more accurate pig detection. Previous studies reported pig tracking [27,28,32,34,38], using index tracking, Kanade–Lucas–Tomasi (KLT) [47], and simple online real-time tracking (SORT) [48] algorithms. The index tracking algorithm calculates the Euclidean distance between consecutive frames to measure the similarity between the two coordinates of a designated object. KLT is a local search that uses gradients, weighted by an approximation of the second derivative of the image. SORT [48] is a practical approach to multiple object tracking, based on rudimentary data relations and state estimation techniques. DeepSORT [41], an extension of SORT, provides simple online and real-time tracking with a deep association metric. In other words, it extends the SORT algorithm to integrate image information based on a deep appearance descriptor. However, utilizing deep learning features in the application particularly slows down the execution speed on embedded boards, which have limited GPU computing power. Therefore, we modified DeepSORT to apply it to embedded boards.

Table 1 summarizes some of the previous methods used for pig detection and/or counting [11,12,13,14,15,16,17,18,19,20,21,22,23,24,25,26,27,28,29,30,31,32,33,34,35,36,37,38]. Detecting individual pigs from a video stream is an essential part of automatically tracking and counting pigs in a hallway. Moreover, it is crucial to meet the real-time requirement to analyze successive video frames without delay, but many previous studies neither report the execution time nor satisfy the real-time requirement. Furthermore, the proposed method should be executed on embedded boards for low-cost monitoring. However, none of them mentioned finding the best trade-off model between accuracy and execution speed on embedded boards. Satisfying both counting accuracy and real-time requirements on embedded boards is very challenging.

## 3. Proposed Method

In this paper, we propose a pig-counting system that automatically calculates the number of pigs that walk through the hallway and are captured by a surveillance camera installed on the wall, allowing the farm workers to check the calculation results. In addition, all tests were performed using a low-cost embedded board, the Jetson Nano, allowing it to be directly applied to pig farms. For counting pigs, we detected individual objects and keep the region of interest (RoI) for detection, exclusive non-RoI. The YOLOv4 and TinyYOLOv4 [40] models are widely used for many object detection applications. In this study, LightYOLOv4, which is a variant and a light-weight version of TinyYOLOv4, is proposed to perform object detection on embedded boards. In addition, a multi-object tracking algorithm was performed based on the detection results to track individual objects. Although the DeepSORT [41] algorithm is widely used in multi-object tracking, in this study, we propose the LightSORT algorithm, which simplifies feature extraction that requires the most computing resources in the DeepSORT algorithm. Finally, the pig-counting system counts individual pigs based on their movement direction in the hallway. The overall system structure of EmbeddedPigCount proposed in this study is shown in Figure 1.

### 3.1. Pig Detection Module

For pig counting, the detection of individual objects should be performed first. In this study, LightYOLOv4, which is lighter than TinyYOLOv4 [40], is proposed for real-time object detection on the Jetson Nano board [39]. Moreover, we apply the TensorRT [49] framework to create GPU-optimized models. We use TensorRT, an inference framework provided by NVIDIA, that reduces execution time by providing optimized model structure for a specific NVIDIA GPU (i.e., an optimized model, which is created by TensorRT, only works on the same GPU model). TensorRT originally works in 32-bit precision, but can also execute models using 16-bit floating point. In this process, we adopt 16-bit floating point which enables faster computation and less memory consumption.

First, the filter clustering (denoted as FC) technique proposed in [23] is applied to shorten the 3 × 3 convolution time, which requires the most computing resources in TinyYOLOv4. The FC method is a pruning technique [50] that reduces the 3 × 3 filter of the convolution layer of the CNN, and multiple filters extracting a similar feature can be grouped into the same cluster. For this clustering, we first prepare 511 features, which can be created with a 3 × 3 binary pattern. Then, each filter in a 3 × 3 convolutional layer is convolved with 511 features and grouped into a cluster with the maximum convolution value. At the end of clustering, some clusters may contain multiple filters. We simply select the filter with the maximum convolution value in each cluster containing multiple filters. The LightYOLOv4 network structure in which the FC method is applied to TinyYOLOv4 is shown in Table 2. Then, we apply TensorRT to obtain a model that performs real-time object detection on the Jetson Nano board.

### 3.2. Pig Tracking Module

In the pig tracking module, we implement object tracking based on the detected objects. Object tracking is generally performed using a Kalman filter. Object tracking using a Kalman filter is a recursive filter that tracks the state of a linear dynamic system, including noise. Currently, the SORT [48] algorithm and the DeepSORT [41] algorithm, which are based on the Kalman filter, are widely used. The SORT algorithm predicts the object position in the next frame (t + 1) by predicting the speed, etc., through the Intersection over Union (IOU) Distance [51] and Hungarian algorithm [52] based on the position of the object appearing in the past (t − 1) and present (t) frames, and compares the predicted result with the actual result to update the position of the object. The SORT algorithm has a fast execution time. However, it produces low accuracy in object tracking. To address this problem, in the DeepSORT algorithm, object tracking is performed using not only the position of the object but also its image features. In this case, the Mahalanobis distance [53] and the cosine distance [54] are used to calculate the similarity of the objects, and CNN is particularly applied to extract the image features of the object. However, because object feature extraction through CNN requires high computing power, it should be simplified to be flawlessly executed on a device with low computing power, such as an embedded board.

Therefore, in this paper, we propose the LightSORT algorithm that executes with high execution speed on an embedded board with nearly no loss of accuracy. Rather than extracting object features using CNN from the DeepSORT [41] algorithm, the size of each object is changed to a size of 10 × 10 and used as an object feature in LightSORT. The 10 × 10 object feature is vectorized to 100 × 1, in the same way as the size of the object feature (128 × 1) in DeepSORT. Figure 2 shows the difference between the existing DeepSORT and the object feature extraction technique of the proposed method, LightSORT.

### 3.3. Pig-Counting Module

Finally, based on the object tracking generated by the LightSORT algorithm, we propose an algorithm to count the number of pigs walking through the hallways of pig farms. The proposed module is executed according to the following rules: (1) If the detected object is a human, an exception is raised so as not to include the detected human in the counting algorithm. (2) On the basis of the counting line (center of the hallway being monitored) in the surveillance camera, the right side is defined as the entrance area, and the left side is the exit area. (3) If the center coordinate of the detected pig’s bounding box is in the entrance area, the state log value is stored as ‘0′. If the center coordinate of the bounding box is in the exit area, the state log value is ‘1′. (4) Moreover, when a pig appears and the given status log value and the current status log value are different, it changes the counting result (CR) value of the entire system. That is, if the pig moved from the entrance area to the exit area, the CR value increased by 1, and if the pig moved the other way around, the CR value decreased by 1. Algorithm 1 shows the pseudo-code written based on the above rules.

Figure 3 shows an example of a pig-counting algorithm. The orange and green zones represent the entrance and exit areas, respectively. If the pig is detected in the entrance area and then disappears after walking through the exit area, the counting for the disappeared pig is complete. Figure 3b depicts a pig moving from the entrance area to the exit area. In this case, the pig was detected in the T-4 frame, and the position of the pig was in the entrance area. The start status and end status are ‘0′. The pig moves to the exit area in the T-1 frame and the end status becomes ‘1′, increasing the CR value by 1. Figure 3c depicts a pig moving from the exit area to the entrance area. The pig was detected in the T-3 frame and the position of the pig was in the exit area. The start and end status are ‘1′. In the T-2 frame, because the end status value is the same as the start value, the CR value does not change. However, in the T-1 and T frames, the CR value decreases by 1 owing to changes between the end status (‘0′) and start status (‘1′). Figure 3d depicts a pig moving back and forth between the entrance area and exit area. First, the pig is detected in the T-5 frame, and because it is in the entrance area, the start status and end status values are stored as ‘0′. In the T-3 frame, the CR value increases by 1 because the end status value is ‘1′. The end status value of the T-2 frame is the same as the start status value, and it does not affect the CR value. Finally, because the end statuses of the T-1 and T frames are ‘1′, the CR value increases by 1.

Figure 4 shows the frame at time T, the log values for individual objects, and the resulting values. The status value is ‘0′, when Pig1 and Pig2 are first detected (start), and current status value is ‘1′ (end). Therefore, the CR value increased by 1. In the case of Pig3, the start status value is ‘1′ and the end status value is ‘0′, thus, the CR value is decreased by 1. However, in the case of Pig4, the start status value and the end status value are the same, therefore, it does not affect the CR value.
**Algorithm 1. EmbeddedPigCount**Input: Video stream from a surveillance cameraOutput: Pig counting result Detect individual pig using LightYOLOv4for (all detected objects):      if object class = person:      continue   if new object:      Add new track to track_list      Save start.status and end.status according to location of the pig   else:      Find a track with prediction results in track_list and connect      Save end.status based on location      Update new location and visualization featurecounting_result = 0 for (all existing tracks):    if start.status = exit area and end.status = entrance area:       counting_result − = 1    if start.status = entrance area and end.status = exit area:       counting_result + = 1 return counting_result 

## 4. Experimental Results

### 4.1. Experimental Environment and Dataset

In this study, the pig-counting model was trained on a PC with an Intel^®^ Core™ i7-9700K CPU @ 3.60GHz, GeForce RTX 2070 GPU, and 32 GB of RAM. To verify that the proposed light-weight counting system can be processed on an embedded board, test experiments were conducted on the Jetson Nano board [39], provided by NVIDIA. The Jetson Nano B01, equipped with a 128-core NVIDIA Maxwell™ GPU, quad-core ARM^®^ A57 CPU, and 4 GB of 64-bit LPDDR4, was used. Hallway images were collected using a Hanwha QNO-6012R [55] surveillance camera, which captured 1920 × 1080 images at 30 frames per second (FPS).

As shown in Figure 5, for object detection in the proposed system, a camera was installed on the wall of a commercial pig farm in Hadong, Gyeongnam, Republic of Korea, and training and test data, which were used in the object detection module, were directly collected from this farm. That is, all the image data were obtained from the pig-moving scenarios scheduled by the commercial pig farm, not from any artificial scenario for this study. Figure 5a shows the actual appearance of the hallway, in which pigs move under the camera surveillance, and Figure 5b shows the captured image that is collected by the camera installed on the wall in the hallway. The surveillance camera recorded the hallway of the pig farm in grayscale (1980 × 1080 resolution) and saved the captured image of objects (pigs, humans) moving in the hallway for training and testing. In addition, to improve object detection accuracy and prevent overfitting, two types of data were added to the training. Figure 5c: a top-down view captured image, showing only pigs, was taken at a pig pen located in Chungbuk, Republic of Korea. Figure 5d: to distinguish people passing by in the hallway from pigs, human data from an open dataset [56] were added to the training. In the cases of Figure 5b,c, the bounding boxes were manually annotated for training. As a result, for the detection module, a total of 2675 images (1702 images of the hallway, 873 images of the pig pen, and 100 images of humans) were used. The pig pen and open dataset were all used as training data, and 1396 images of the hallway were assigned for training, and the remainder of the hallway data (306 images) was used as test data (see Table 3).

To measure the tracking and counting accuracy of the proposed system, video clips were collected from the same camera, as shown in Figure 5a,b. The video clips contained at least one action of the pig and the person moving in the hallway (left, right, right, or left). The bounding box, class, and track ID information were manually added to the 3035 objects to test the tracking accuracy. In addition, 130 video clips, ranging from 10 s (s) to 300 s, were collected to test the counting accuracy, with a minimum of one pig and a maximum of 34 pigs appearing in the collected video clips. In the final collected video clips, unnecessary parts, other than the hallway, were removed through image preprocessing, thereby constructing a dataset that could focus only on moving objects in the hallway. In this application, through Real-Time Streaming Protocol (RTSP) [57] streaming audio/video data from an IP camera server, the Jetson Nano embedded board receives the video in real time. A video sample is shown in Figure 6.

### 4.2. Results with Pig Detection Module

First, we examined the accuracy of object detection in the hallway, using LightYOLOv4. The hallway image was input with a size of 320 × 320, the learning rate was 0.001, the optimizer was SGD, the momentum was 0.949, and 200 epochs of training were conducted. Table 4 shows the accuracy of the object detection in the hallway. When only hallway data were used, the mAP was 94.95%; however, the experimental result with the addition of the pig pen data and the open dataset was 96.95%, which confirms the increase in accuracy by 2%. Then, we measured the execution time of LightYOLOv4. We obtained 38.8 FPS for detection and, thus, the whole counting system can satisfy the real-time processing speed on the Jetson Nano board.

The detection accuracy and execution time of this experiment are important factors that can significantly affect the pig-tracking and counting results. Despite the use of LightYOLOv4, which is a light-weight model with fast processing speed, but relatively low accuracy, it shows a reasonable detection accuracy of this experimental result, indicating that LightYOLOv4 can be applied to pig counting on the Jetson Nano board.

### 4.3. Results with Pig Tracking Module

As described in Section 3.2, in the proposed system, the DeepSORT [41] algorithm is not used as is, but the image feature extraction is modified using LightSORT for faster speed and smooth GPU allocation. DeepSORT and LightSORT were written by Python. Moreover, libraries, such as NumPy, sklearn-learn, OpenCV, VidGear, were used, and detailed parameters, such as init, maximum age, and max distance were set to 3, 20, and 0.9, respectively, to conduct this experiment. Both pig-tracking modules were tested with Ubuntu 18.04 on an NVIDIA Jetson Nano embedded board.

Table 5 compares the tracking accuracy of the DeepSORT [41] and LightSORT algorithms. Multiple Object Tracking Accuracy (MOTA) [58], which measures the overall accuracy of both tracker and detection, is often used in multi-objective object tracking (MOT). Therefore, MOTA and ID switch (IDsw), which measures the number of times that the ID assigned to an object changes, are used as measuring indicators. The experiment was conducted with 3000 frames of video images to measure the tracking accuracy. In the case of MOTA, DeepSORT performed more accurately than LightSORT; however, the difference was insignificant at 0.03%. By contrast, the IDsw result shows a smaller number of ID switches in LightSORT than in DeepSORT. In summary, this implies that there is little difference in accuracy between DeepSORT and LightSORT. This indicates that the image feature used in DeepSORT is helpful in distinguishing objects with different appearances; however, in this experimental environment, the visual difference between pigs is not notable, thus, the image feature from DeepSORT is not beneficial for the pig-counting system. Therefore, the results of this experiment show that the LightSORT method, which extracts light-weight image features, rather than DeepSORT, which extracts precise image visual features, is a suitable tracking algorithm for this system, in terms of execution time and accuracy.

For a video-based real-time system, the execution speed of the entire system must exceed 30 FPS to process the video received at 30 FPS through the surveillance camera in real time, without delay. This means that the execution speed of the entire counting system, including modules, such as detection and counting, in addition to tracking, must exceed 30 FPS. Figure 7 shows the difference in execution time between DeepSORT and LightSORT, according to the number of objects (pigs + humans) appearing in one frame on the Jetson Nano board. When one object appears in DeepSORT, the execution speed is 22.62 ms, which is equivalent to 44.2 FPS (the value shown in red color is FPS); however, the execution time increases significantly as the number of objects increases. For example, when 10 objects appear in one frame, it drastically drops to 5.4 FPS, which is much below the real-time standard. Considering that 10 or more objects frequently appear in one frame in the actual hallway of pig farms, DeepSORT is not practically suitable for counting systems on the Jetson Nano board. By contrast, LightSORT shows a fast execution speed of 0.12 ms (8623 FPS) for one object and 0.46 ms (2152 FPS) for 10 objects. Even if other additional algorithms of the counting system are executed simultaneously, a stable execution speed is guaranteed on LightSORT. These results demonstrate the suitability of the proposed LightSORT over DeepSORT for this system, in terms of performance speed.

### 4.4. Results with Pig-Counting Module

To evaluate the proposed counting system, 130 hallway video clips were collected and used to verify its accuracy. The counting accuracy was measured based on the number of pigs that moved from right to left in the image from the video clips. The right area of the image corresponds to the entrance area, and the left area corresponds to the exit area in the proposed system. For example, if 10 pigs moved from right to left in a video clip and two of those pigs moved back from left to right, the final count was eight (=10 − 2). The detailed results for the 130 video clips are shown in Table 6. Each video clip contained a minimum of one and a maximum of 34 pigs. Accuracy is calculated as the number of pigs correctly detected (denoted as *Nc*) compared to the number of pigs in the video clips (denoted as *N*), and the number of pigs correctly detected is calculated through the difference between the ground truth and the actual counting result. Therefore, this system provides 99.44% counting accuracy (711 pigs were correctly counted out of 715 pigs appearing in 130 video clips). That is, it shows that the proposed system can count flawlessly, even in a complex situation, in which 10 or more pigs appear.

Figure 8 shows the qualitative results of the proposed system. Each image is a captured image of the counting result video. A pig is annotated by a yellow box, and a person is annotated by a red box. The number in white on the bottom left corner of each captured image represents the counting result of the pig’s movement, from the start of the video to the current time (the number of pigs passing through the counting line from right to left). Counting ground truth (counting GT) is the correct answer for the videos. If the number at the bottom left in the last frame matches the counting GT, it is the correct answer. Otherwise, this is an incorrect answer. As a result, Figure 8a–c shows the correct pig-counting results, whereas Figure 8d shows the incorrect pig-counting results in this experiment (video clip ‘130′).

More specifically, Figure 8a shows a video of typical pig movement in the hallway, and the results of the six pigs migrating from right to left are correctly counted. All pigs were detected and tracked properly, and when a pig crossed the counting line, the counting number was accurately updated. In the case of Figure 8b, it is the result of the pigs moving from left to right instead of right to left. Two pigs moved from left to right, and as a result, the counted number shows ‘−2′ correctly. Figure 8b shows that the proposed counting system can count pigs moving not only from right to left, which is the purpose of this system, but also in both directions, where users initialize it. This shows the flexibility of the proposed counting system. Furthermore, Figure 8c shows the result of a situation in which 23 pigs appear in the video clip. This video contained the largest number of pigs, except for the video clip ‘130′, which recorded an extreme condition. Despite the situation where many pigs appeared, accurate counting was performed without difficulty. This outcome shows that the proposed system can operate robustly, even in the case of overlapping problems in a dense environment, which may occur when pigs are moving. Figure 8d depicts a situation in which pigs are moving through the overcrowded hallway. As shown in the second image in Figure 8d, pig counting is executed properly, even in the dense state, but in the third image, when a pig rides over another pig, several pigs lying underneath are excluded from detection and tracking. In this process, an error occurs in counting and, as a result, the pig count is incorrectly checked as ‘−3′ (the correct counting number is 0). If the correct answer to pig counting is 0, pigs appear from the right side of the screen, remain on the left side for some period, and then all the pigs move back to the right again. As shown in Figure 8d, plenty of pigs were induced to move in the opposite direction of the original movement direction and, thus, the pigs collided in the hallway, resulting in an indistinguishable overlap between pigs. As a result, we identified a limit that could cause a counting error for a severe overlapping situation. However, despite the occurrence of various overlapping situations in many video clips, the achievement of high accuracy for 715 pigs shows the robustness of the proposed system.

Table 7 indicates the accuracy for 130 video clips and shows a counting accuracy (99.44%). Note that the counting accuracy is higher than that of detection and tracking. It shows that the efficient counting algorithm of this system, which focuses detection and tracking results near the counting line, provides robust pig counting that is not highly dependent on detection and tracking accuracy. This result shows that it is possible to perform pig counting that is robust, even in the hallways of actual pig farms, where many variables can occur. In addition, the overall FPS of the pig-counting system proposed in this study was measured in the Jetson Nano board environment, and the average FPS of 130 video clips is presented. The measurement result shows a processing speed of 30.6 FPS, which means that hallway images can be handled in real time, with 99.44% counting accuracy on the Jetson Nano board.

Furthermore, the actual execution time for each module is shown in Table 8. It shows that the execution time of the detection and tracking module occupies more than 90% (78.9% + 12.5% respectively) of the total execution time, despite the weight reduction. It implies that the weight reduction of the detection and tracking module in this study can be an effective contribution for building an embedded system (in the case of the execution time of the tracking module, additional time more than the result of Section 4.3 is required due to the process of storing, deleting, and uploading the complex tracking information of multiple objects for a certain period of time). The experiments consequentially verified that the proposed system can count pigs in the actual hallways of pig farms, and the counting system can be applied and operated flawlessly, using the embedded board without financial burden on the pig farms.

## 5. Conclusions

Real-time pig counting is particularly important for facilitating efficient management in large-scale pig farms. However, applying pig counting on embedded boards poses challenges, such as satisfying both accuracy and real-time requirements simultaneously.

In this study, a light-weight method was proposed for pig counting, using a low-cost embedded board. First, we reduced the computational cost of the TinyYOLOv4 object detector, without losing detection accuracy, by applying the filter clustering technique. Then, we modified the DeepSORT object tracker to reduce the computational cost of feature extraction without losing tracking accuracy. A combination of these light-weight detection and tracking methods, with an accurate counting algorithm, can achieve high accuracy with real-time speed, even with some pigs passing through the counting zone back and forth. Based on the experiment with 130 video clips obtained from a pig farm, our light-weight deep-learning-based method, EmbeddedPigCount, can achieve acceptable counting accuracy, with real-time speed on a USD 100 NVIDIA Jetson Nano embedded board; thus, efficient farm management is possible in a cost-effective manner.

In addition, for more real-world video monitoring setups, specialized for livestock farms, we plan to attach a camera module to an embedded board and check whether semi-supervised learning (at nighttime), as well as testing (at daytime), on the embedded board is possible in harsh livestock monitoring environments as our future study. If such continual learning and testing on an embedded board are feasible in a stand-alone form for low monetary cost (i.e., the total estimated cost is less than USD 500), we believe that it will be a practical solution to the problem of unseen data in large-scale farms, as well as in others, such as cow and poultry farms.

## Figures and Tables

**Figure 1 sensors-22-02689-f001:**
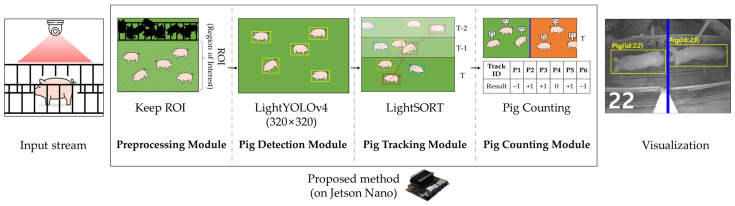
Overview of EmbeddedPigCount.

**Figure 2 sensors-22-02689-f002:**
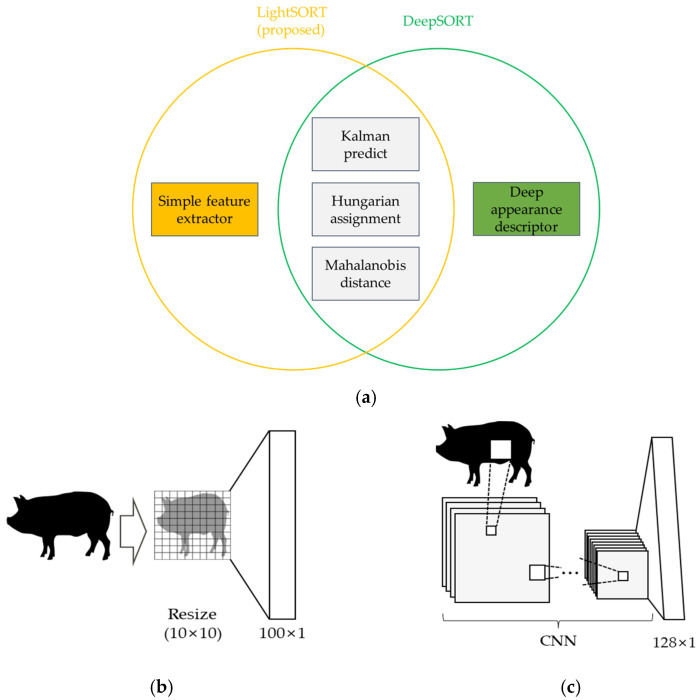
Difference in feature extraction between DeepSORT [41] and LightSORT. (**a**) A Venn diagram showing each SORT algorithm. (**b**) A simple feature extractor algorithm while extracting object features in the LightSORT. (**c**) A deep appearance descriptor while extracting object features in the DeepSORT.

**Figure 3 sensors-22-02689-f003:**
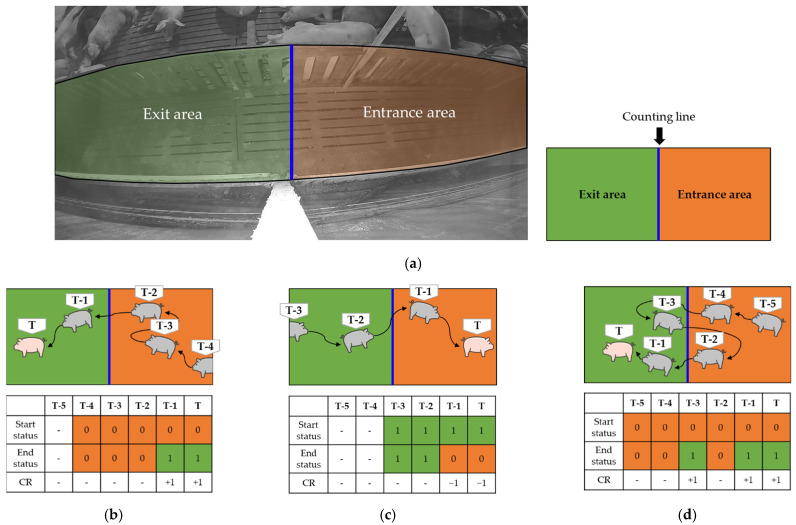
Illustration of the hallway pig-counting algorithm. (**a**) An actual pig farm to which the system proposed in this paper is applied. (**b**) An example video clip of a pig moving from an entrance area to an exit area. (**c**) An example video clip of a pig moving from an exit area to an entrance area. (**d**) An example video clip of a pig moving from an entrance area to an exit area, with an intermediate back and forth.

**Figure 4 sensors-22-02689-f004:**
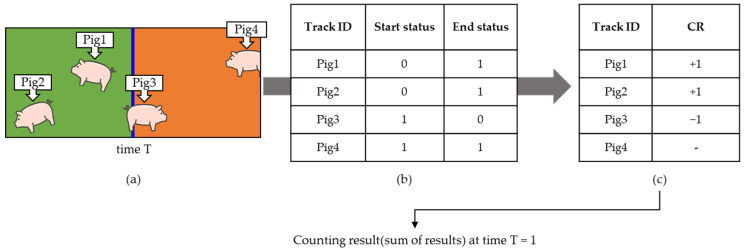
Illustrative example that may occur at time T with the hallway pig-counting algorithm. (**a**) Input frame. (**b**) The status log value corresponding to each object. Track ID shows the name corresponding to each object, the start status shows the status value when the object was first created, and End status shows the value corresponding to the current (T). (**c**) The result values affecting the CR corresponding to individual objects.

**Figure 5 sensors-22-02689-f005:**
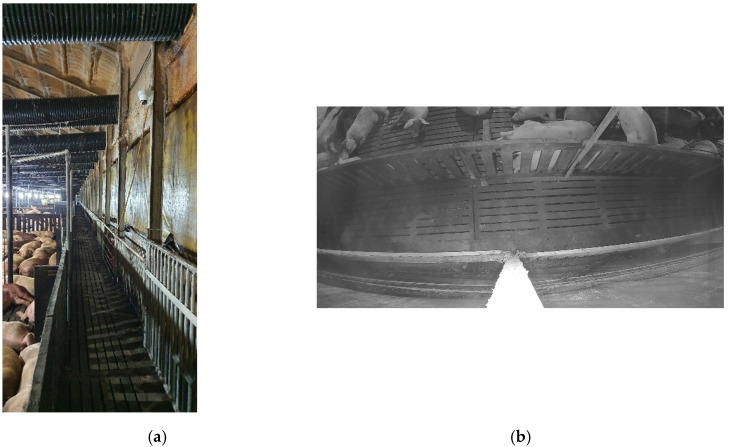
Example training and test datasets. (**a**) Hallway in real environment. (**b**) Still shot from a surveillance camera installed on the wall in the hallway. (**c**) Pig pen data sample. (**d**) Open dataset of people in top view.

**Figure 6 sensors-22-02689-f006:**
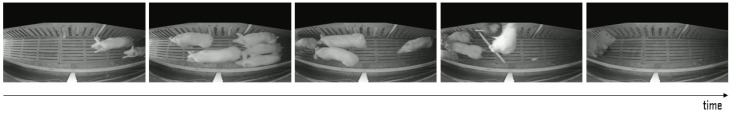
Video clip samples for pig counting.

**Figure 7 sensors-22-02689-f007:**
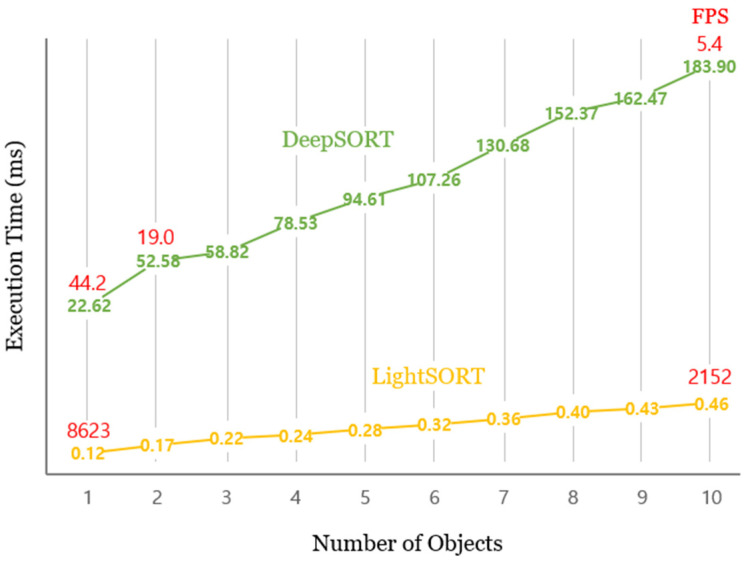
Speed comparison between DeepSORT [41] and LightSORT on Jetson Nano, with a variable number of objects in a frame.

**Figure 8 sensors-22-02689-f008:**
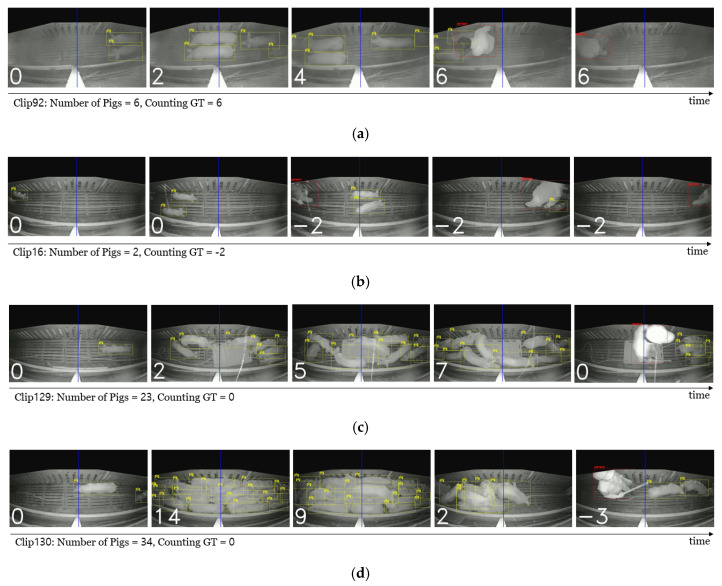
Qualitative results of the proposed counting system. The Counting GT (Ground Truth) represents the number of pigs that moved from right (i.e., entrance area) to left (i.e., exit area). (**a**) Counting in common situations. (**b**) Counting in the opposite direction. (**c**) Counting in dense situations. (**d**) Incorrect counting result.

**Table 1 sensors-22-02689-t001:** Some of the pig detection and/or counting results (published during 2012–2021).

Application	Data Type	Algorithm	No. of Pigs in Each Image/Video	Execution Time per Image (ms)	Target Platform	Reference
Pig Detection	Image	Image Processing	9	Not Specified	PC	[11]
		Image Processing	7~13	Not Specified	PC	[12]
		Image Processing	1	Not Specified	PC	[13]
		Image Processing	Not Specified	500	PC	[14]
		Image Processing	Not Specified	Not Specified	PC	[15]
		Image Processing	13	2	PC	[16]
		Image Processing	Not Specified	1000	PC	[17]
		Deep Learning	1	Not Specified	PC	[18]
		Deep Learning	Not Specified	500	PC	[19]
		Deep Learning	∼32	142	PC	[20]
		Image Processing	4	921	PC	[21]
		Deep Learning	6	500	PC	[22]
		Image Processing + Deep Learning	9	29	Embedded Board	[23]
		Deep Learning	~79	Not Specified	PC	[24]
		Deep Learning	13	41~2000	PC	[25]
		Image Processing + Deep Learning	9	~190	Embedded Board	[26]
	Video	Image Processing	22	Not Specified	PC	[27]
		Image Processing	1	Not Specified	PC	[28]
		Image Processing	22	Not Specified	PC	[29]
		Image Processing	17~20	Not Specified	PC	[30]
		Deep Learning	1	50	PC	[31]
		Deep Learning	4	Not Specified	PC	[32]
		Deep Learning	20	250	PC	[33]
Pig Counting	Image	Image Processing	8	Not Specified	Not Specified	[34]
		Image Processing	9	Not Specified	Not Specified	[35]
		Deep Learning	~40	42	PC	[36]
	Video	Deep Learning	~250	313	Embedded Board	[37]
		Deep Learning	~18	Not Specified	Not Specified	[38]
		Deep Learning	~34	32	Embedded Board	Proposed

**Table 2 sensors-22-02689-t002:** LightYOLOv4 network architecture.

#	Layer	Filters	Size/Stride	Input	Output
0	Convolutional	27	3 × 3/2	320 × 320 × 1	160 × 160 × 27
1	Convolutional	49	3 × 3/2	160 × 160 × 27	80 × 80 × 49
2	Convolutional	45	3 × 3/1	80 × 80 × 49	80 × 80 × 45
3	Route 2				
4	Convolutional	31	3 × 3/1	80 × 80 × 22	80 × 80 × 31
5	Convolutional	28	3 × 3/1	80 × 80 × 31	80 × 80 × 28
6	Route 4, 5				
7	Convolutional	64	1 × 1/1	80 × 80 × 59	80 × 80 × 64
8	Route 2, 7				
9	Maxpool		2 × 2/2	80 × 80 × 109	40 × 40 × 109
10	Convolutional	86	3 × 3/1	40 × 40 × 109	40 × 40 × 86
11	Route 10				
12	Convolutional	56	3 × 3/1	40 × 40 × 43	40 × 40 × 56
13	Convolutional	47	3 × 3/1	40 × 40 × 56	40 × 40 × 47
14	Route 12, 13				
15	Convolutional	128	1 × 1/1	40 × 40 × 128	40 × 40 × 128
16	Route 10, 15				
17	Maxpool		2 × 2/2	40 × 40 × 214	20 × 20 × 214
18	Convolutional	164	3 × 3/1	20 × 20 × 214	20 × 20 × 164
19	Route 18				
20	Convolutional	83	3 × 3/1	20 × 20 × 82	20 × 20 × 83
21	Convolutional	83	3 × 3/1	20 × 20 × 83	20 × 20 × 83
22	Route 20, 21				
23	Convolutional	256	1 × 1/1	20 × 20 × 166	20 × 20 × 256
24	Route 18, 23				
25	Maxpool		2 × 2/2	20 × 20 × 420	10 × 10 × 420
26	Convolutional	189	3 × 3/1	10 × 10 × 420	10 × 10 × 189
27	Convolutional	256	1 × 1/1	10 × 10 × 189	10 × 10 × 256
28	Convolutional	174	3 × 3/1	10 × 10 × 256	10 × 10 × 174
29	Convolutional	18	1 × 1/1	10 × 10 × 174	10 × 10 × 18
30	YOLO output				
31	Route 27				
32	Convolutional	128	1 × 1/1	10 × 10 × 256	10 × 10 × 128
33	Upsample		/2	10 × 10 × 128	20 × 20 × 128
34	Route 23, 33				
35	Convolutional	120	3 × 3/1	20 × 20 × 384	20 × 20 × 120
36	Convolutional	18	1 × 1/1	20 × 20 × 256	20 × 20 × 18
37	YOLO output				

**Table 3 sensors-22-02689-t003:** Datasets used for training and testing.

Annotation	Dataset	Class	Train Set (Images)	Test Set (Images)
Manually annotated	Hallway	Pig + Human	1396	306
	Pig pen	Pig	873	-
Open dataset	people in top view [57]	Human	100	-

**Table 4 sensors-22-02689-t004:** Accuracy comparison of LightYOLOv4 with various data configurations.

	Only Hallway Data	Hallway + Pig Pen + Open
Person (AP)	94.96%	98.50%
Pig (AP)	94.94%	95.39%
mAP	94.95%	96.95%

**Table 5 sensors-22-02689-t005:** Accuracy comparison between DeepSORT [41] and LightSORT.

	MOTA↑	IDsw↓
DeepSORT [41]	89.88%	25
LightSORT	89.85%	22

**Table 6 sensors-22-02689-t006:** Counting results in detail for each clip.

	Number of Pigs (*N*)	Counting Results		
		GroundTruth	EmbeddedPigCount	Number of Pigs Correctly Counted (*Nc*)
Clip01~06	1	−1	−1	1
Clip07~08	1	0	0	1
Clip09~15	1	1	1	1
Clip16	2	−2	−2	2
Clip17	2	0	0	2
Clip18~25	2	2	2	2
Clip26	3	2	2	3
Clip27~37	3	3	3	3
Clip38	4	3	3	4
Clip39~59	4	4	4	4
Clip60	5	−5	−5	5
Clip61	5	1	1	5
Clip62	5	3	3	5
Clip63	5	4	4	5
Clip64~84	5	5	5	5
Clip85~86	6	−2	−2	6
Clip87	6	0	0	6
Clip88~91	6	5	5	6
Clip92~110	6	6	6	6
Clip111~114	7	7	7	7
Clip115	8	5	5	8
Clip116	8	6	6	8
Clip117~118	8	8	8	8
Clip119	9	−8	−8	9
Clip120	9	−1	−1	9
Clip121	9	7	7	9
Clip122	10	−1	−1	10
Clip123	12	0	0	12
Clip124	14	4	4	14
Clip125~126	21	0	0	21
Clip127	23	0	1	22
Clip128 Clip129	23 30	0 0	0 0	23 30
Clip130 Total	34 715	0 -	−3 -	31 711

**Table 7 sensors-22-02689-t007:** Accuracy and FPS of the proposed counting system.

	Accuracy	FPS (Avg.)
Proposed EmbeddedPigCount	99.44%	30.6

**Table 8 sensors-22-02689-t008:** Execution time of each module of the proposed counting system.

	Execution Time (Milliseconds)	Proportion (%)
Detection module Tracking module	25.8 4.1	78.9 12.5
Etc.	2.7	8.3
Total	32.7	100

## Data Availability

Not applicable.

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
