# Peer review of "EmbeddedPigCount: Pig Counting with Video Object Detection and Tracking on an Embedded Board"

_sensors, 2022, doi:10.3390/s22072689_

Round 1
Reviewer 1 Report
In this paper, the authors propose a on-device pig counting pipeline. Although the novelty is marginal, the pipeline seems to work in the real world. In the meanwhile, the accuracy seems to be comparable with the state-of-the-art performance. The paper is drafted well and straightforward to follow. Given to those, this work is still valuable from a practical point of view.
Author Response
We would like to thank you for the meticulous and useful comments. We hereby mention that the constructive criticism has greatly contributed to enhancing level of completion of this paper. We will describe comprehensive answer in authors’ position about the point of each reviewer one by one faithfully.

Reviewer 2 Report
The method proposed in this paper is interesting, and the reviewer believes that the results of the performance evaluation based on actual data will be meaningful to many readers. As for the quality of the English text, there are some ambiguities (e.g., the mixture of "lightweight" and "light-weight"), and it is recommended to have the text checked by native English speakers.
Author Response

(The authors gave the same response as above.)

Reviewer 3 Report
In this paper, the authors describe a pig counting method based on a low-cost device and an adapted deep learning method. The device and the method take into account the particular conditions related to the specific environment in which they have to be implemented.
After a general introduction that recalls the issues that exist in this particular field and the technological choices related to it, the authors present the context of their study. In this part, the authors establish a state of the art of the different detection, tracking and counting techniques already implemented and evaluate them with respect to the particular needs related to the study.
After this presentation, the authors present in detail their method and their algorithm which is based on three steps:
1. detection (which computational efficiency is improved by a filter clustering technique without loss of specificity)
2. monitoring,
3. counting.
In a section entitled "Experimental results", the paper focuses on a detailed description of the means implemented (camera, embedded system, networks and numerical methods used) as well as the data sets used. The experimental results are then presented with a convincing evaluation of the different steps mentioned.
Finally, the authors conclude on the efficiency of their approach and validate the choices made, in particular from the point of view of the lightness of the detection method.
Although it applies to a very specific domain, the content of this paper is quite original and new. The content and the contributions are sufficient and the results are rather well presented. The approach remains very technical and concerns a very specific community. However, we can estimate that the results obtained and the methods implemented could be applied to other fields. It would be interesting to identify and mention them in the conclusion.
Without representing a major contribution, the work presented in this publication is of good quality and well presented. However, I recommend that the layout be reworked to avoid tables being cut off and presented on two separate pages.
In fact, from my point of view, there is no reason why this article should not be published in the journal Sensors.
Author Response

(The authors gave the same response as above.)

Reviewer 4 Report
The authors presented an interesting work on pig counting by edge computing. This certainly has a good contribution in the field of livestock production. Specifically, an algorithm was developed for detection and tracking of pigs from video clips. The authors were able to apply some improvements in order to use the algorithm on an embedded device. Albeit these improvements were minimal, it was found to be an adequate contribution since optimizing an algorithm for edge computing is quite challenging. The paper is quite well-written but there were some missing details that will be very important for readers that are not familiar with this kind of topic. Here are some specific concerns that have to be addressed:
- L11: “do not take a pose for counting” kindly improve this expression
- Table 1: The first row cell “learning” might be a mistake
- L138-L153: This part was rather confusing. At first, the authors mentioned they used “TinyYOLOv4” but then the network architecture is for the “LightYOLOv4”. Does this mean that the LightYOLOv4 is a variation of the TinyYOLOv4? If yes, then please also clarify this in L123-126.
- L152: Other readers might not be too familiar about TensorRT. Please describe it in detail and explain how it was used for optimizing the model used.
- In relation to 4, it might be good to know which programming language was used to implement the entire algorithm (since we all know that different programming languages may affect the processing time in different ways).
- Figure 2: This diagram will look very confusing for other readers. Please try to improve this to make sure that the difference between the two algorithms will be very clear.
- Figure 2: How was the size of 10x10 determined?
- Figure 3. Please be consistent between “enter zone” or “entry zone”.
- L236: Please fix this table/figure and follow a format appropriate for pseudo-codes
- L258-L265: Perhaps put the training/validation/testing dataset information in a table to avoid confusion.
- Figure 6: What kind of camera module was used for collecting these video clips? Was it also the surveillance camera? If that is the case, how will then this be replicated when the Jetson Nano is used? Was the surveillance camera connected to the Jetson Nano during video collection?
- L291-293: It could be very interesting for the readers to know about the computation time required for each component of the algorithm. This will also help convince readers about the contribution of this work such as modifying and improving the existing algorithms for edge computing application.
- L306: inits?
- L313: Please mention the complete definition of MOTA
- L334-L340: Did the authors do any other optimization methods other than modifying the algorithms? There was also no mention which libraries or programming languages were used for the models, which may also affect the computation time of the algorithm. Thus, it is also worthy to mention which OS the Jetson Nano was operating on.
- L334: Why did the algorithm need to exceed 30fps in the first place?
- L339-341: This was confusing. What does the (8,623FPS) mean?
- Figure 8: Provide the definition of GT
- How much is the approximate total cost of the system? Considering that Jetson Nano is quite expensive, will this be practical for real-world applications? How do the authors then propose it will be in a real-world setup? Please mention this as a recommendation to the readers.
- Test experiments were applied using Jetson Nano. However, did the authors consider that it might also be hot inside a pig farm? It is quite a concern since this will cause the Jetson Nano to throttle and affect its processing speed.
- Rather than repeating about the testing results such as the “99.44% counting accuracy”, please provide more conclusions that will help readers understand what were the contributions of this work and how it will be applied in real-world applications.
Author Response

(The authors gave the same response as above.)
